# Tea Polyphenols Prevent and Intervene in COVID-19 through Intestinal Microbiota

**DOI:** 10.3390/foods11040506

**Published:** 2022-02-10

**Authors:** Qiao Xiang, Lu Cheng, Ruilin Zhang, Yanan Liu, Zufang Wu, Xin Zhang

**Affiliations:** 1Department of Food Science and Engineering, Ningbo University, Ningbo 315211, China; xq10723@163.com (Q.X.); liuyanan@nbu.edu.cn (Y.L.); wzfwpf@163.com (Z.W.); 2Department of Food Science, Rutgers, The State University of New Jersey, New Brunswick, NJ 08901, USA; lc894@scarletmail.rutgers.edu; 3Key Laboratory of Animal Protein Deep Processing Technology of Zhejiang Province, Ningbo University, Ningbo 315211, China

**Keywords:** COVID-19, intestinal microecology, tea polyphenols, microbiota

## Abstract

Although all countries have taken corresponding measures, the coronavirus disease 2019 (COVID-19) is still ravaging the world. To consolidate the existing anti-epidemic results and further strengthen the prevention and control measures against the new coronavirus, we are now actively pioneering a novel research idea of regulating the intestinal microbiota through tea polyphenols for reference. Although studies have long revealed the regulatory effect of tea polyphenols on the intestinal microbiota to various gastrointestinal inflammations, little is known about the prevention and intervention of COVID-19. This review summarizes the possible mechanism of the influence of tea polyphenols on COVID-19 mediated by the intestinal microbiota. In this review, the latest studies of tea polyphenols exhibiting their own antibacterial and anti-inflammatory activities and protective effects on the intestinal mucosal barrier are combed through and summarized. Among them, (−)-epigallocatechin-3-gallate (EGCG), one of the main monomers of catechins, may be activated as nuclear factor erythroid 2 p45-related factor 2 (Nrf2). The agent inhibits the expression of ACE2 (a cellular receptor for SARS-CoV-2) and TMPRSS2 to inhibit SARS-CoV-2 infection, inhibiting the life cycle of SARS-CoV-2. Thus, preliminary reasoning and judgments have been made about the possible mechanism of the effect of tea polyphenols on the COVID-19 control and prevention mediated by the microbiota. These results may be of great significance to the future exploration of specialized research in this field.

## 1. Introduction

COVID-19 remains an important public health issue of great concern at home and abroad, and it is still in the state of a global pandemic [1]. During the normalization stage of the prevention and control of the epidemic of COVID-19, China effectively controlled the outbreak and spread of the epidemic by adopting the strategy of “import of external prevention and rebound of internal prevention” and effectively reduced the occurrence of death cases [2]. The social economy recovered quickly, and various measures were highly recognized by the public, and the positive trend of the epidemic continued to consolidate [3]. Although the spread of the local epidemic has been stopped, the international epidemic continues to rise rapidly, and the pressure of “imported prevention and control” in China continues to increase at present [2].

The most common clinical symptoms of patients with new coronary pneumonia are fever, dry cough, and dyspnea, but gastrointestinal symptoms cannot be ignored [4]. The survey shows that patients with gastrointestinal symptoms progress more rapidly than those without gastrointestinal symptoms, that the proportion of severe and critical illness is large, and that the survival prognosis is poor [5]. This situation can be altered through a reasonable diet and other methods. Among them, the role of tea polyphenols in regulating intestinal microbiota and maintaining gastrointestinal homeostasis has been confirmed by a large number of studies [6]. This article intends to establish the internal connection between tea polyphenols and COVID-19 by regulating the intestinal microbiota and to explore the possibility of tea polyphenols to regulate the intestinal microbiota to prevent and control COVID-19.

In this environment, this article aims to seek a new method for the prevention and intervention of COVID-19, that is, the utilization of tea polyphenols to regulate the intestinal microbiota [7,8]. The regulation of intestinal microbiota by tea polyphenols is essential to maintain intestinal homeostasis and prevent bacterial infections [9]. This method has previously been used in research in the fields of immunity, aging, and cardiovascular and metabolic diseases and has achieved certain positive results [10]. For this reason, it is reasonable to use tea polyphenols to regulate intestinal microecology for the prevention and intervention of COVID-19 [11]. Therefore, this review introduces COVID-19 and tea polyphenols, and the possible mechanisms by which tea polyphenols regulate intestinal microbiota and influence COVID-19 under the mediation of intestinal microbiota.

## 2. The Infection Mechanism and Prevention of COVID-19

Under such a circumstance where the novel crown epidemic has lasted for more than one year, people’s understanding of the genus of coronaviruses, including the novel crown virus, has reached a new level [12]. Coronavirus is a type of positive single-stranded linear RNA virus that envelops the envelope [13]. It is widely distributed in poultry, humans, and other mammals [14] and belongs to the order of the *Mantlevirus*, Coronaviridae, and Orthocoronavirinae subfamily. It is the largest known RNA virus of humans [15]. This mature virus virion diameter is about 60 to 220 nm and looks like a coronavirus under the electron microscope. Because of that, it is called a coronavirus. Coronavirus poses a serious threat to the health of humans and animals [16]. It generally causes diseases of the respiratory tract, gastrointestinal tract, and central nervous system of humans and animals, which threaten human health and cause economic losses [17]. Most coronavirus strains have a narrow host range, but zoonotic viruses tend to invade new hosts [18]. With the continuous overlap of human and wild animal ecosystems, the possibility of new coronavirus appearing means that COVID-19 has been the result of rampant infection and mutation, which can be described as fierce [19].

COVID-19 has brought great challenges and threats to human life [20]. The causative agent of COVID-19 is a new type of β-coronavirus (CoVs), which is 80% and 50% similar to severe acute respiratory syndrome coronavirus (SARS-CoV) and Middle East respiratory syndrome virus (MERS-CoV), respectively, and named SARS-CoV-2 [21,22,23]. Most patients infected with the new coronavirus have poor body heat, cough, fatigue, anorexia, and greasy moss as their main symptoms [24,25].

Once a virus infects a cell, it first penetrates the cell membrane of the host cell, and then un-shells, synthesizes and assembles macromolecules. Finally, the progeny virions are released outside the cells. Prior to this, the first step in initiating infection is the specific binding of the viral envelope protein to the viral receptor on the cell surface, allowing the virus to be adsorbed on the cell surface [26]. According to research, coronaviruses have obvious tissue tropism and host specificity. This is because the spike protein S or HE protein of the virus envelope specifically binds to host cell receptors and mediates membrane fusion [27]. SARS-CoV-2 is the pathogen of COVID-19, which broke out in 2019 and eventually became globally prevalent [28,29]. The most closely related chrysanthemum bat coronavirus bat-SL-CoVZC45 has a homology of about 88%, which is much greater than SARS-CoV (79%) and MERS-CoV (50%) [30]. But in terms of structure, SARS-CoV-2 possesses a receptor binding domain similar to SARS-CoV, even though there are some differences in some key amino acid residues. Therefore, the infection mechanism of SARS-CoV-2 and SARS-CoV may have certain similarities [31,32]. The SARS-CoV of the β-coronavirus mainly uses the spike protein S to bind to the ACE2 of the alveolar and small intestinal epithelial cells. At the same time, it binds to another receptor, CD209L, to form a complex, which uses caveolin and clathrin to invade cells through independent endocytosis and initiate infection [32]. Later studies have indeed confirmed that SARS-CoV-2 recognizes the receptor on the cell membrane, namely ACE2, through its spike protein, and enters the cell through endocytosis under the mediation of the receptor. Then new virus particles are replicated and released to infect other host cells (Figure 1) [33].

COVID-19 mainly affects the lungs, causing interstitial pneumonia and respiratory distress syndrome, and meanwhile, it also affects other organs [34,35]. Most patients with new coronary pneumonia suffer from other diseases. In the treatment of this type of patient, it is easy to cause complications, eventually resulting in death from multiple organ failure, etc., rather than the new coronavirus infection itself [36]. Many official platforms have issued different prevention and treatment recommendations for close contacts, children, ordinary adults, and special populations, such as traditional Chinese medicine and virus vaccines. But in summary, the focus of the prevention and intervention of the new crown pneumonia virus is to strengthen basic exercises, pay attention to personal hygiene, control the source of infection, cut off the transmission route, and maintain a precautionary mentality to stay in awe of life [37]. At present, vaccination is mostly used at home and abroad, although the principles of the new crown vaccine produced by different manufacturers are different [38]. For example, China’s new crown vaccine is an inactivated vaccine. It refers to cultivating the corpse of the new coronary pneumonia virus in vitro and then injecting it into the body to produce antibodies. It can be maintained for up to six months and requires repeated vaccination. Inactivated vaccine induces IgG-based milk-derived immunity to play a role in the intestine, which is systemic and supports intestinal mucosal sIgA. Pfizer’s new coronavirus vaccine is a nucleic acid vaccine. It is to encode related protein genes directly into the human body to stimulate the human body to produce antibodies, which is the most effective and technical method. However, the immune regulation and effector mechanisms of the body may lead to the destruction of antigen-expressing cells, resulting in the release of intracellular antigens and the activation of autoimmunity, which is one of its potential hazards. The new Russian crown vaccine is a modified adenovirus used as a vector to inject the protein gene of the new crown virus. After it is injected into the human body, it stimulates the production of antibodies [39]. There is no doubt that, regardless of whether the principles are the same, different vaccines are scientifically proven, and safety and effectiveness are the top priority [40].

The gut microbiota influences the development and function of the immune system, which in turn regulates gut microbiota diversity. An effective vaccine should be able to elicit a protective immune response against a given viral preparation, while gut microbiota composition and diversity directly or indirectly modulate the immune response to the vaccine. A healthy gut microbiota is a key factor in maintaining gut homeostasis, which is critical for optimal vaccine performance. Nutrient imbalances and gut dysbiosis negatively impact host health, immunity, and gut barrier, and their synergistic interactions may affect vaccine efficacy.

With the spread of the new coronavirus around the world, variant strains are also emerging. Among them, the Delta strain has quickly become the dominant strain due to its strong transmission, high viral load, and strong pathogenicity. It has brought new challenges to the global epidemic prevention and control. The current study found that the new coronavirus Delta strain has a higher viral load than Beta and other variant strains, and the lower cycle threshold and longer virus release period of this strain significantly enhanced its transmissibility when tested by polymerase chain reaction (PCR). The Delta strain showed higher replication efficiency in airway organoids and the human airway epithelial system, and the spike-affinity-increasing ACE2 receptor protein enhanced viral adhesion and made it easier to enter the body. The spike protein is the “key” that the virus uses to penetrate the door of human cells, and it is also the target of most vaccines. It is precisely because of these mutant characteristics of the Omicron variant that it increases the risk of secondary infection of the human body with the new coronavirus. Some experts believe that because AIDS will weaken the human body’s immunity, the variant of the Omicron virus is more likely to infect people with weakened immunity, which in turn can cause other diseases, which can be called the “AIDS” of the new coronavirus. Therefore, stricter protection and control strategies are needed, and new and effective prevention and control methods are more actively developed.

## 3. The Interaction of Plant Polyphenols with Intestinal Microbiota on the Host

The intestinal microbiota is an important part of the physiological functions of animal body digestion, metabolism, and vitamin synthesis [41]. With the development of basic and clinical research, the research on the relationship between the intestinal microbiota and the pathogenesis of autoimmune diseases has been gradually deepened, and the influence of the complex network of intestinal microbiota composition and metabolites on the disease will be further understood. In addition to digestion in the body, the intestinal microbiota could produce some enzymes to promote the decomposition of polysaccharides, plant polyphenols, and the production of some vitamins, which helps the host’s intestinal health [42]. The structure, composition and number of the intestinal microbiota are variable. Therefore, prevention and treatment strategies accurately altering the intestinal microbiota are worthy of research; the intestinal microbiota can be controlled by targeting to restore the intestinal microecological balance, and finally achieve the purpose of disease prevention, control, and treatment.

Plant polyphenols, also known as plant tannins, are secondary metabolites with polyphenolic hydroxyl structure [43]. They are commonly found in vegetables, fruits, Chinese herbal medicines, and plant seeds. They are especially abundant in tea, coffee, red grapes, kidney beans, and red wine. Many in vitro, animal, and disease case studies have shown that plant polyphenols can promote the growth of probiotics, selectively inhibit the growth of pathogenic bacteria, optimize the structure of the intestinal microbiota, and adjust the intestinal microecological balance [44]. It is more than that polyphenols may affect various metabolic or signaling pathways, ultimately affecting human diseases such as cardiovascular disease, various inflammations, and cancers, which always are caused by viruses invading the human body [45]. In addition, the intestinal microbiota can metabolize high-molecular-weight plant polyphenols into more bioactive metabolites to improve their bioavailability [46].

The metabolism and absorption of plant polyphenols are inseparable from the participation of intestinal microbiota, and their metabolites can exert more beneficial effects on the intestinal microbiota and the body [47]. Wang Jing et al. found that *Lactobacillus plantarum ZLP001* could strengthen the intestinal barrier by enhancing the defense function of intestinal epithelial cells and regulating the intestinal microbiota [48]. The research results showed that polyphenols have obvious protective effects on the intestinal tract: maintaining the balance of the number of *Escherichia coli*, *Lactobacillus*, and *Enterococcus* in the intestine [49]. Other research found that polyphenols can promote intestinal motility in mice, inhibit diarrhea caused by senna, and have a good role in treating gastrointestinal disorders and regulating gastrointestinal balance.

Studies have reported that rosemary extract has a significant inhibitory effect on inflammation [50]. As an anti-inflammatory dietary supplement, it can improve the inflammatory symptoms of the colon. Additionally, in the rat foot swelling experiment, it can significantly reduce the expression of inflammatory factors in the body. Studies have shown that the anti-inflammatory activity of rosemary is closely related to its rich compounds such as carnosic acid, carnosol, and ursolic acid. Furthermore, in lipopolysaccharide (LPS)-induced inflammatory cell models, it can well inhibit the production of inflammatory factors.

EGCG, a catechin monomer, is the main component of green tea polyphenols. It is a polyphenol compound that has been studied extensively [51]. It has antibacterial, antiviral, antioxidant, anti-inflammatory, and anti-tumor effects, and so on. Epidemiological studies have confirmed that green tea is effective in preventing cancer and cardiovascular and nervous system diseases and has strong antioxidant capacity. EGCG can inhibit a variety of inflammatory enzymes and cytokines, such as COX2, IL-6, and IL-1β induced and secreted by TNF-α; its anti-inflammatory ability is positively correlated with its intake of total phenols [52].

The 5NLRP3 inflammasome is the core of the inflammatory response. The main mechanisms of EGCG inhibiting inflammation are:EGCG can inhibit the initiation and assembly activation process of NLRP3 inflammasome;As an inhibitor of NLRP3 inflammasome activation, EGCG inhibits the LPS initiation phase and assembly activation pathway in macrophages, mitigates cell scorching, and inhibits NLRP3 inflammasome activation by blocking the spatial location of mitochondrial translocation and ASC speck formation during inflammasome activation;EGCG can improve the activation level of inflammasomes in mouse-derived macrophages induced by a high-fat diet.

In a clinical trial of catechin antiviral efficacy, it was found that, when taking catechin/theanine capsules for 5 months, the probability of clinical influenza infection was significantly lower than that of the placebo control group, and the time for patients to get rid of clinically confirmed infection was also shorter, significantly shorter than the control group, suggesting that taking catechin/theanine can effectively prevent influenza virus infection. Studies have confirmed that tea polyphenols can inhibit the secretion of hepatitis B virus surface antigen (HBs Ag) and hepatitis B virus e antigen (HBe Ag) and can significantly reduce the expression of HBV-DNA in the supernatant of the cell culture system, indicating that tea polyphenols have the potential to be developed into an anti-HBV drug. In clinical trials, EGCG showed a good effect in reducing HIV infection, and the polyphenols were well tolerated without adverse reactions.

Moreover, plant polyphenols can fight tumors by influencing tumor cell signal transduction, inducing tumor cell differentiation and apoptosis, regulating related enzyme activities and tumor cell cycle, and inhibiting tumor cell proliferation. Through the PI3K-AKT signaling pathway, EGCG do COX-2wn-regulates cyclooxygenase-2 (COX-2), activates caspase-3 and caspase-9, induces the apoptosis of liver cancer cells, inhibits the activity of matrix metalloproteinase (MMP)-2 and MMP-9, promotes the apoptosis of B-lymphoma cells, downregulates the expression levels of PI3K and AKT/NF-κB, and makes the liver cancer cell SMMC7721 stagnate in the S phase [53].

Research found that under the synergistic effect of quercetin, resveratrol, ethanol, and grape polyphenols, polyphenols can significantly inhibit nitric oxide synthase, which plays an important role in protecting human health [54]. Others found through in vitro experiments that Lingzhi capsules (RM) and Tegeen (tea polyphenol content >98%) greatly reduce the number of human breast cancer cells MDA-MB-31, proving that they inhibit the proliferation and differentiation of cancer cells, and other aspects have significant synergistic effects. Based on the results of a large amount of data, the mechanisms of cooperative anti-cancer could be come down to removal of free radicals and carcinogenic factors, regulation of enzyme activity in the body, and protection of DNA, etc. [55].

The dietary intervention of ingesting foods high in plant polyphenols in an appropriate amount and time can regulate the intestinal microbiota to a certain extent [56]. For example, eating vegetables and fruits helps keep the body healthy, possibly due to the strong antioxidant activity of polyphenols [57]. In addition, polyphenols can reshape the intestinal microbiota to enhance host–microbe interaction [58]. The composition of intestinal microflora affects the bioavailability of polyphenols and their metabolites [59]. The regulation of intestinal microflora can increase the metabolism of sugars and improve intestinal barrier function and energy consumption, and it also reduces inflammation, insulin resistance, obesity, weight gain, and dyslipidemia. These improvements ultimately help reduce metabolic diseases, gastrointestinal inflammation, and related complications.

## 4. Possible Mechanisms of Intestinal Microbiota Regulating COVID-19

Studies on the “microbiota–gut–brain axis” show that intestinal microbes affect the body’s immune system and nervous system, thereby altering brain functions and interfering with the pathological development of the central nervous system (CNS). Environmental factors such as an unbalanced diet, abuse of antibiotics, and lifestyle changes can change the composition of the intestinal microbiota and even lead to their collapse, leading to increased intestinal permeability, blood–brain barrier permeability, and inflammation of the peripheral and central nervous system, and ultimately cause neurological diseases [60]. Therefore, the regulation of the gut microbiota through long-term individualized dietary modification is a viable strategy for the prevention and intervention of COVID-19. Tea polyphenols can be one of the links.

## 5. The Regulating Effect of Tea Polyphenols on Intestinal Microecology

The various medical functions of tea have been widely reported, and some studies have explored the regulatory effects of tea on intestinal microbiota. Currently, the research directions mainly focus on two aspects; one is the inhibitory effect on the intestinal dysfunctional microbiota, and the other is the growth-promoting effect on beneficial microorganisms. Some studies have shown that black tea extracts and the intake of tightly pressed tea juice can increase trypsin activities, thus activating proteases in the small intestine and stomach to accelerate the decomposition and absorption of food [61]. The main mechanism is that black tea can regulate the structure of the intestinal microbiota, tending to diversity, increase the number of probiotic lactic acid bacteria such as Enterococcus, Lactobacillus, and Bifidobacterium and reduce the number of conditionally pathogenic bacteria such as *Escherichia coli*, thus keeping the intestinal micro-ecosystem healthy and stable.

Tea polyphenols have a positive regulatory effect on the animal intestinal mucosal barrier [62]. The intestinal mucosal barrier, as a defense system against external infections plays an important role in maintaining intestinal homeostasis, balancing intestinal microbiota, and body health [63]. TLR2-mediated signaling can avoid the apoptosis of intestinal epithelial cells, slow down intestinal mucosal damage, and maintain the integrity of intestinal barrier function [64]. One study found that tea polyphenol pretreatment significantly inhibits the downregulation of TLR2 m RNA expression in cells induced by ETEC K88 attack, suggesting that tea polyphenols might maintain the integrity of intestinal epithelial cell barrier function through TLR2-mediated signaling. It has also been shown that EGCG reduces damage to gastric mucosal cells caused by *Helicobacter pylori* infection and promotes the regeneration of gastric epithelial cells [65]. Lee et al. found that green tea polyphenol extracts ameliorate ethanol-induced acute gastric mucosal injury by reducing the ethanol-induced elevated expression of cyclooxygenase-2 and inducible nitric oxide synthase in gastric mucosa coincidentally. It works by reducing the increase in the expression of cyclooxygenase-2 and inducible nitric oxide synthase in the gastric mucosa caused by ethanol to inhibit the transient redox-sensitive transcription factors NF-κB, AP-1, and mitogen-activated protein kinase activation [66]. At present, the regulatory effects of tea polyphenols on the intestinal mucosal barrier have been proven, and tea polyphenols have a more prominent performance on the intestinal microbiota [67]. However, the current oral bioavailability of tea polyphenols is low, and there is little use of tea polyphenols in improving intestinal disorders and maintaining intestinal health. Therefore, the specific regulatory mechanism and signal transduction pathways of tea polyphenols on the intestinal mucosal barrier still need to be further research.

At the same time, tea polyphenols have significant antibacterial and anti-inflammatory effects. The main mechanism of tea polyphenols exerting anti-inflammatory effects is to regulate the expression of inflammatory factors to achieve the purpose of inhibiting inflammation [68]. In the experiment, scholars used mice with lung injury as the research object. During the experiment, they were given tea polyphenol extracts and compared to those untreated tea polyphenol extracts. Scholars found that the infiltration of neutrophils, tumor necrosis factor, nitrite, and other substances in the tissues are significantly reduced, which fully showed that tea polyphenols have a good therapeutic effect on lung injury caused by acute inflammation and can significantly reduce the permeability of granulocytes in the tissue, tumor necrosis factor, nitrite, and other substances. These results indicated that the use of tea polyphenols have a good therapeutic effect on lung injury caused by an acute inflammatory effect. In addition, some researchers studied liver inflammation in alcoholic liver injury and found that the use of tea polyphenols is very effective in reducing serum alanine aminotransferase levels and reducing polymorphonuclear leukocyte infiltration, which would be a good preventive control for liver injury caused by acute inflammation. Tumor necrosis factor-alpha (TNF-α) is a pro-inflammatory cytokine that promotes the production of various chemokines and cytokines that initiate the acute and chronic phases of inflammation. It has shown that TNF-α may cause NF-κB activation and F-actin rearrangement, thereby disrupting tight junction disruption causing barrier dysfunction [69]. The anti-inflammatory effect of tea polyphenols was further confirmed by Oz et al. They found that small amounts of green tea polyphenols and EGCG reduce TNF-α levels in the blood of mice with colitis [70]. Moreover, GTP has a significant inhibitory effect on acute colitis induced by glycolic anhydrides, which is dependent on the inhibition of inflammatory factors [71]. In a study, EGCG was found to reduce the overproduction of IL-6 and IL-8 by 50% and 60%, respectively, in a small intestinal epithelial cell inflammation model but had no effect on their mRNA levels, presumably regulating inflammatory genes through a post-transcriptional regulatory mechanism [72]. It is also found that green tea extract effectively alleviated the inflammatory response in IL-2-deficient mice, inhibited the overproduction of TNF-α and IFN-γ, and reduced the incidence of severe colitis. The results are in consistence with an in vitro experiment with IEC-6 small intestinal epithelial cells. Other effects of green tea polyphenols (GTP) on intestinal inflammation have been identified in other studies, such as the ability of GTP to effectively reduce chemotherapeutic drug-induced enterotoxicity and decrease macrophage inflammatory protein-2 and myeloperoxidase activity [73]. Several of the above studies have confirmed that GTP has a clear inhibitory effect on intestinal inflammation.

In addition, as an activator of Nrf2, EGCG can inhibit the entry of SARS-CoV-2 into host cells and provide host cells with preparations against SARS-CoV-2 infection [74,75]. Nrf2, the cytoprotective transcription factor, regulates the expression of a wide array of genes involved in antioxidation, detoxification, inflammation, immunity, and antiviral responses [76]. Decreased Nrf2 in differentiated human nasal epithelial cells will increase virus entry and replication, while Nrf2 activators, such as EGCG and sulforaphane, will decrease virus entry and replication [75]. Many studies have also shown that genetic and pharmacological manipulations activating the Nrf2 pathway can inhibit virus replication and prevent virus-induced oxidative damage and inflammation [77]. In addition, by activating Nrf-2 regulated heme oxygenase 1, EGCG can mediate antiviral response by increasing the expression of type 1 interferon and mitigating SARS-CoV-2. At the same time, the inflammatory response is initiated by the interference of Nrf2 and NF-κB in inflammatory tissues, where innate immune cells are recruited (Figure 2) [78,79]. EGCG interferes with SARS-CoV-2 spike-receptor interaction and blocks the entry of pseudotyped-SARS-CoV-2 lentiviral vectors with an IC50 value of 2.5 μg/mL [80]. Studies in silico suggest that EGCG may also inhibit papain-like protease protein and prevent COVID-19 by binding to its S1 ubiquitin binding site. It is shown that EGCG can be used as a broad-spectrum therapy for asymptomatic and symptomatic COVID-19 patients [81].

Tea regulates intestinal microbiota via indirect and direct approaches: the indirect one is through the inhibition of various dysbiosis microorganisms in the intestinal tract; tea polyphenols and tea pigments and other active ingredients have a significant inhibitory effect on a variety of dysbiosis microorganisms, while the direct effect is that tea polyphenols and other active ingredients to protect and promote the growth of beneficial intestinal microbiota regulate the intestinal microecological balance. Studies have confirmed that the astringent component of tea, catechin, has a bactericidal effect on enterohemorrhagic *Escherichia coli* O157:H7. *E. coli* can cause diarrhea, damage mucosal epithelial cells, and invade mucosal epithelium to form local ulcers and inflammation when it is dysregulated [82]. Tea polyphenols are recognized by scholars in many countries as a broad-spectrum, potent, and low-toxicity antibacterial drug [83]. Tea polyphenols have inhibitory effects on many pathogenic bacteria, such as *Bacillus dysenteriae* and *Escherichia coli*, which have certain curative effects on dysentery. In addition, tea polyphenols in tea combine with proteins playing the role of tannin protein, thus relieving intestinal tension and calming intestinal peristalsis to achieve anti-inflammatory and anti-diarrheal effects. The addition of tea polyphenols can inhibit the growth of bacteria and prolong the shelf life of feed [84]. Meanwhile, it is found that the addition of tea polyphenols to broiler diets could significantly reduce the number of lactobacilli and play a positive role in regulating the gastrointestinal microbiota [85]. Studies have found that tea polyphenols have inhibitory effects on Gram-negative bacteria, *Proteus vulgaris*, *Staphylococcus aureus*, Lactobacillus, and intestinal pathogenic bacteria [86]. Taking advantage of the antibacterial effect of tea polyphenols, spraying a certain concentration of tea polyphenol solution on fresh fruits or vegetables can exhibit antiseptic effects and freshness preservation [87]. At the same time, scientific researchers have proven that tea polyphenols have a positive effect on poultry diseases, including avian influenza and coccidiosis [88].

Gut microbiota can transform tea polyphenols into corresponding metabolites through different metabolic pathways, while these metabolites can exert a modulatory effect on intestinal microbiota by selective prebiotic and antibacterial activities [89]. After oolong tea polyphenols intervention, a corresponding decrease in the Firmicutes/Bacteroidetes ratio was observed [90]. It has been reported that constituents found in all major tea types, predominantly L-theanine, polyphenols, and polyphenol metabolites, are capable of functioning through multiple pathways simultaneously to reduce the risk of depression [91]. Our previous study provided a global view that tea polyphenols from oolong tea might alleviate the circadian rhythm disorder of the host, contributing to micro-ecology improvement [92]. Therefore, the interaction of tea polyphenols with intestinal microbiota and their implication for COVID-19 are of great interest [93].

## 6. The Impact of Intestinal Homeostasis on COVID-19

According to reports, the first patient with new coronary pneumonia in the United States experienced gastrointestinal symptoms such as diarrhea, nausea, and vomiting, and abdominal discomfort during the course of the disease [94]. It has been confirmed that SARS-CoV-2 virus can directly enter cells using angiotensin-converting enzyme II (ACE2) as a receptor [95]. ACE2 is rarely expressed in the esophageal epithelium, but is abundantly expressed in the cytoplasm of glandular epithelial cilia and stomach and intestinal epithelial cells. This means that SARS-CoV-2 can directly enter host cells, especially stomach and intestinal epithelial cells, causing inflammation. The high expression of ACE2 in the small intestine makes small intestinal epithelial cells highly susceptible to SARS-CoV-2 infection [96]. The interaction between SARS-CoV-2 and ACE2 may disrupt the function of ACE2, leading to intestinal malabsorption, stimulating the enteric nervous system, and causing diarrhea [97]. Wang et al., among 651 patients with new coronary pneumonia in Zhejiang Province, found that 22.97% of patients with new coronary pneumonia with gastrointestinal symptoms were severely or critically ill [98]. The rate was significantly higher than that of patients without gastrointestinal symptoms. It can be seen that patients with gastrointestinal symptoms progress more rapidly than those without gastrointestinal symptoms, that the proportion of severe and critical illness is large, and that the survival prognosis is poor.

With in-depth research on the clinical symptoms, signs, and pathological changes of coronavirus disease, it has been gradually realized that the new coronavirus not only causes lung tissue damage but also involves multiple tissues and organs throughout the body, especially the gastrointestinal tract and immune system. It has been found that the imbalance of gut microbiota is associated with more and more diseases, from gastrointestinal diseases (such as inflammatory bowel disease) to metabolic disorders [99]. In the past decade, a large number of animal and clinical studies have shown that intestinal microbiota not only mediates the physiological processes of host metabolism and immunity but also plays an important role in the two-way response between gastrointestinal tract and CNS [100]. Under normal physiological conditions, on the one hand, intestinal microorganisms can interact with the host through a complex dynamic network regulation mechanism, participate in regulating the metabolism and immunity of the host, and protect the integrity of the intestinal mucosal barrier structure; on the other hand, beneficial bacteria can interact with the host. It produces antibacterial substances that are more beneficial to the body and compete to seize nutrients and seize favorable sites to resist the invasion of pathogens [101]. When the steady-state balance of the intestinal microecology is disrupted, beneficial bacteria will decrease, and harmful bacteria will increase, the intestinal barrier will be damaged, and the immune system function will be destroyed, which will trigger a variety of diseases and threaten human health [102]. Many scholars have found that intestinal microbes play an important role in regulating the body’s immunity, improving the microenvironment of the intestinal microbiota, preventing the migration of the microbiota, and protecting the mucosal barrier of the gastrointestinal tract [103]. Changes in the intestinal microbiota are associated with local inflammation, and systemic Inflammation is closely related, and its role in COVID-19 has attracted attention [104].

The intestinal microecological balance is closely related to various diseases such as digestive system diseases and immune diseases. Tryptophan and its metabolite nicotinamide are key regulators of the intestinal microbiota and inflammation [105]. The ACE2 on the luminal surface of the small intestinal epithelial cells binds to the amino acid carrier B0AT1, so that the dietary tryptophan of the intestinal microbiota is absorbed [106]. Nicotinamide and dietary tryptophan can stimulate the mTOR pathway to affect the intestinal antimicrobial peptides [107]. SARS-CoV-2 receptor ACE2 can regulate the dynamic balance of intestinal microbes through amino acids and affect the expression of antimicrobial peptides [108]. Studies have shown that SARS-CoV-2 can interact with ACE2 on the surface of host cells and reduce the expression of ACE2 through endocytosis [109]. Experiments have shown that severe colitis in ACE2-deficient mice is caused by impaired tryptophan uptake. Lack of ACE2 will cause severe damage to local tryptophan homeostasis and reduced production of antimicrobial peptides, thereby changing the susceptibility to intestinal inflammation [110]. Therefore, SARS-CoV-2 may affect the absorption of tryptophan through ACE2, leading to the reduction of antimicrobial peptides, thereby changing the intestinal microflora and promoting intestinal inflammation.

It is worth noting that there are many beneficial ways to regulate the imbalance of the intestinal microbiota, including fecal microbiota transplantation (FMT), probiotics application, and dietary intervention-using non-digestible dietary substrates such as prebiotics [111]. In dietary intervention, EGCG has been shown many redox and specific inhibitory activities, in cell lines and rodent models. As mentioned above, it may be applicable for the prevention and intervention of COVID-19. EGCG may inhibit the expression of ACE2 and TMPRSS2 on the cell surface by activating Nrf2, thereby inhibiting SARS-CoV-2 infection. EGCG may also inhibit SARS-CoV-2 Mpro, a protease necessary for virus reproduction. The direct and indirect antioxidant activity of EGCG may prevent the oxidative stress induced by SARS-CoV-2. EGCG can reduce endoplasmic reticulum stress and the life cycle of SARS-CoV-2 by inhibiting the activity and expression of GRP78 resident in the endoplasmic reticulum. EGCG can also prevent the ALI/ARDS, thrombosis, sepsis, and pulmonary fibrosis associated with cytokine storm. Overall, EGCG has been shown to prevent SARS-CoV-2 infection, inhibit the life cycle of SARS-CoV-2, and curb the cytokine storm, oxidative stress, ER stress, thrombosis, and sepsis caused by SARS-CoV-2 and pulmonary fibrosis [112]. Therefore, in theory, tea consumption can play a role as a means of dietary intervention to prevent and control new coronary pneumonia and to adjust the intestinal microbiota according to the structural and functional characteristics of EGCG.

In general, regulating the intestinal microbiota can still play a role in the prevention of COVID-19 to some extent. On the one hand, patients with COVID-19 have intestinal microbiota disorders and impaired barrier function and may have gastrointestinal symptoms, such as nausea, vomiting, and diarrhea, and gastrointestinal symptoms are the first manifestation of their symptoms [113]. On the other hand, intestinal microbiota can become a turning point in the prognosis of the disease by regulating the local and systemic immune systems of the intestine (Figure 3) [114]. Therefore, restoring and adjusting the balance of intestinal microbiota is helpful to the prevention of COVID-19. With the in-depth exploration of the intestinal microbiota related to the COVID-19, the advantages of Chinese and Western medicine complement each other, and new targets and new ideas for the treatment of the new type of coronavirus pneumonia are found from the perspective of regulating the micro-ecology. The combined efforts of Chinese and Western medicine have given full play to their respective advantages in the treatment of new coronavirus pneumonia [115].

## 7. Conclusions

Evidence has indicated that intestinal microbes play a key role in the prevention and intervention of COVID-19. At the same time, the role of tea polyphenols in regulating intestinal microbiota has been confirmed by many studies, and we inferred that we can try to use tea polyphenols as a starting point to mediate the intestinal microbiota to prevent and intervene regarding COVID-19. At present, there is still room for development in the prevention and intervention of COVID-19 by regulating intestinal microbes. The intervention of tea polyphenols to improve the imbalance of intestinal microbes provides a new target for the prevention intervention of COVID-19. The regulation of intestinal flora by tea polyphenols can also create the advantage of intestinal homeostasis for optimal vaccine performance. However, compared with vaccines, polyphenol interventions are not targeted and only universal. It is hoped that through the study of specific dietary intervention mechanisms, the occurrence of COVID-19 virus infections can be reduced or prevented.

## Figures and Tables

**Figure 1 foods-11-00506-f001:**
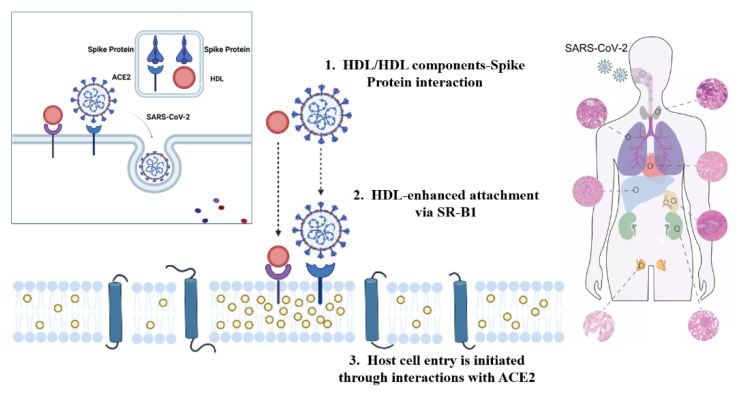
COVID-19 pathology mechanism.

**Figure 2 foods-11-00506-f002:**
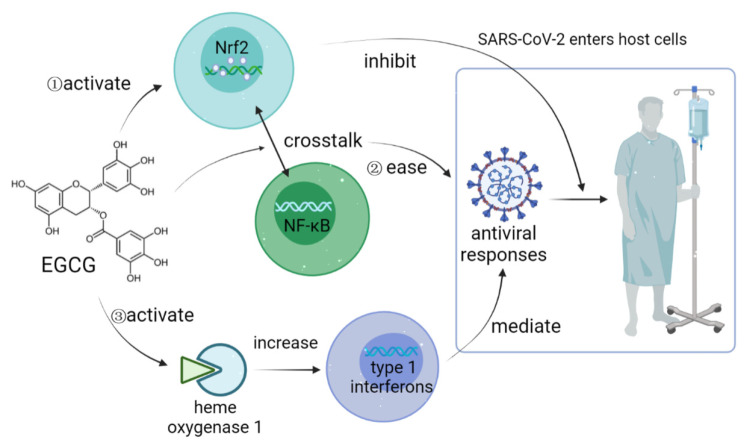
EGCG acts as a Nrf2 activator against SARS-CoV-2 infection.

**Figure 3 foods-11-00506-f003:**
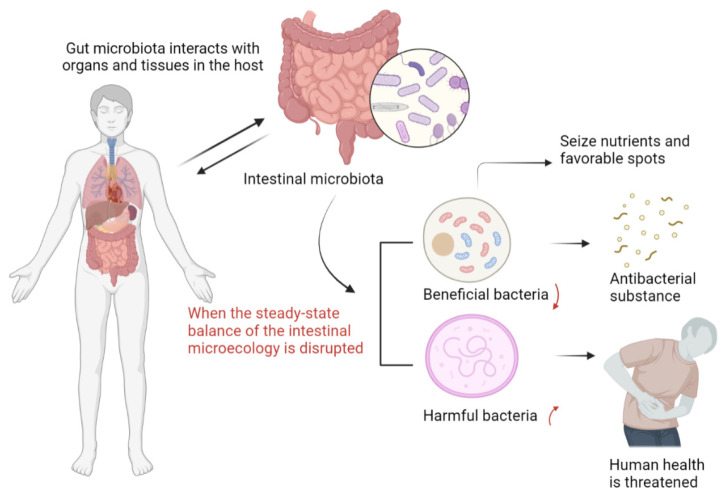
The direct interaction between the intestinal flora and the human body.

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
