# Peer review of "Tea Polyphenols Prevent and Intervene in COVID-19 through Intestinal Microbiota"

_foods, 2022, doi:10.3390/foods11040506_

Round 1

Reviewer 1 Report

Authors Xiang and the team reviewed the impact of Tea polyphenols mediated COVID-19 by the intestinal microbiota. Considering the current pandemic situation and the impact in the world health system and economy, this review article give other aspect of the issue.

The review structure and the date collection are well done.

The following points should be addressed.

  • The interaction of the plant compounds with the possible immune cell types could be addressed in detail (Page 4).
  • Bacterial genus and species names are denoted in italics and non-italics. Example page 4, line 159; Page 5, line 236.
  • The latest mutation of the virus can be addressed.
  • Impact of various COVID vaccine on gut microbiota can be addressed.
  • Figures are not self-explainable (Figure 1-3).

Author Response

Responds to the reviewer's comments:

Dear Reviewers:

Thank you for your letter and for the reviewer's comments concerning our manuscript entitled “Tea polyphenols prevent and intervene in COVID-19 through intestinal microbiota” (ID: Foods-1560245). Those comments are very helpful for revising and improving our paper. We have studied comments carefully and have made a correction which we hope meet with approval. Revised portions are marked by “Track Changes”. The main corrections in the paper and the responses to the reviewer's comments are as follows:

Reviewer #1: Authors Xiang and the team reviewed the impact of Tea polyphenols mediated COVID-19 by the intestinal microbiota. Considering the current pandemic situation and the impact in the world health system and economy, this review article give other aspect of the issue.

The review structure and the date collection are well done.

The following points should be addressed.

The interaction of the plant compounds with the possible immune cell types could be addressed in detail (Page 4).

Response: According to your proposal, a new section of the article has been added about the antiviral effects of plant polyphenols.

“In the clinical trial of catechin antiviral efficacy, it was found that taking catechin/theanine capsules for 5 months, the probability of clinical influenza infection was significantly lower than that of the placebo control group, and the time for patients to get rid of clinically confirmed infection was also shorter. Significantly shorter than the control group, suggesting that taking catechin/theanine can effectively prevent influenza virus infection. Studies have confirmed that tea polyphenols can inhibit the secretion of hepatitis B virus surface antigen (HBs Ag) and hepatitis B virus e antigen (HBe Ag), and can significantly reduce the expression of HBV-DNA in the supernatant of the cell culture system, indicating that tea polyphenols Has the potential to be developed into an anti-HBV drug. In clinical trials, EGCG showed a good effect of reducing HIV infection, and the polyphenols were well tolerated without adverse reactions.”

Bacterial genus and species names are denoted in italics and non-italics. Example page 4, line 159; Page 5, line 236.

Response: According to your suggestion, the errors in the text have been corrected, We've highlighted them in red.

The latest mutation of the virus can be addressed.

According to your proposal, a new section of the article has been added about the new coronavirus Delta strain.

“With the spread of the new coronavirus around the world, variant strains are also emerging. Among them, the Delta strain has quickly become the dominant strain due to its strong transmission, high viral load, and strong pathogenicity. It has brought new challenges to the global epidemic prevention and control. The current study found that the new coronavirus Delta strain has a higher viral load than Beta and other variant strains, and The lower cycle threshold and longer virus release period of this strain significantly enhanced its transmissibility when tested by Polymerase Chain Reaction (PCR). And the Delta strain showed higher replication efficiency in airway organoids and the human airway epithelial system, and the spike-affinity-increasing ACE2 receptor protein enhanced viral adhesion and made it easier to enter the body. The spike protein is the "key" that the virus uses to penetrate the door of human cells, and it is also the target of most vaccines. It is precisely because of these mutant characteristics of the Omicron variant that it increases the risk of secondary infection of the human body with the new coronavirus. Some experts believe that because AIDS will weaken the human body's immunity, the variant of the Omikron virus is more likely to infect people with weakened immunity, which in turn can cause other diseases, which can be called the "AIDS" of the new coronavirus. Therefore, stricter protection and control strategies are needed, and new and effective prevention and control methods are more actively developed.”

Impact of various COVID vaccine on gut microbiota can be addressed.

Response: We appreciate your suggestions and recommendations. We have carefully referred to other relevant articles and made modifications in our paper. In addition, a section was added to bridge the relationship between vaccines and intestinal flora.

The gut microbiota influences the development and function of the immune system, which in turn regulates gut microbiota diversity. An effective vaccine should be able to elicit a protective immune response against a given viral preparation, while gut microbiota composition and diversity directly or indirectly modulate the immune response to the vaccine. A healthy gut microbiota is a key factor in maintaining gut homeostasis, which is critical for optimal vaccine performance. Nutrient imbalances and gut dysbiosis negatively impact host health, immunity, and gut barrier, and their synergistic interactions may affect vaccine efficacy.”

Figures are not self-explainable (Figure 1-3).
Response: Thank you very much for your suggestions, some textual additions have been replaced all three images.

Reviewer 2 Report

The aim of this article is to explain the possible mechanism of the influence of tea polyphenols on the COVID -19 mediated by the microbiota. 
This narrative review is well written, appropriately structured, and has a sufficient number of references.
In the various sections, the authors have explained the mechanism of infection and prevention of COVID -19, the interaction of plant polyphenols with the gut microbiota on the host, possible mechanisms of intestinal microbiota COVID -19, the regulatory effect of tea polyphenols on intestinal microecology, and the effects of intestinal homeostasis on COVID -19. 
In addition to this structure, I recommend stating the strengths and limitations of the proposed approach. 
Also, please correct the Latin names of the microbial species - they should be written in italics.

Author Response

Responds to the reviewer's comments:

Dear Reviewers:

Thank you for your letter and for the reviewer's comments concerning our manuscript entitled “Tea polyphenols prevent and intervene in COVID-19 through intestinal microbiota” (ID: Foods-1560245). Those comments are very helpful for revising and improving our paper. We have studied comments carefully and have made a correction which we hope meet with approval. Revised portions are marked by “Track Changes”. The main corrections in the paper and the responses to the reviewer's comments are as follows:

Reviewer #2: The aim of this article is to explain the possible mechanism of the influence of tea polyphenols on the COVID -19 mediated by the mic robiota.

This narrative review is well written, appropriately structured, and has a sufficient number of references.

In the various sections, the authors have explained the mechanism of infection and prevention of COVID -19, the interaction of plant polyphenols with the gut microbiota on the host, possible mechanisms of intestinal microbiota COVID -19, the regulatory effect of tea polyphenols on intestinal microecology, and the effects of intestinal homeostasis on COVID -19.

In addition to this structure, I recommend stating the strengths and limitations of the proposed approach.

Also, please correct the Latin names of the microbial species - they should be written in italics.

Response: Thank you very much for your suggestion, which provide another new perspective for our review and discussion. We try our best to learn and master. In light of your advice on the strengths and limitations of the proposed method, we have added some additions to the "Conclusion" section. 

Evidences indicated that intestinal microbes play a key role in the prevention and intervention of COVID-19. At the same time, the role of tea polyphenols in regulating intestinal microbiota has been confirmed by many studies, and we inferred that we can try to use tea polyphenols as a starting point to mediate the intestinal microbiota to prevent and intervene COVID-19. At present, there is still large room for development in the prevention and intervention of COVID-19 by regulating intestinal microbes. The intervention of tea polyphenols to improve the imbalance of intestinal microbes provides a new target for the prevention intervene of COVID-19. The regulation of intestinal flora by tea polyphenols can also create the advantage of intestinal homeostasis for optimal vaccine performance. However, compared with vaccines, polyphenol interventions are not targeted and only universal. It is hoped that through the study of specific dietary intervention mechanisms, the occurrence of COVID-19 virus infections can be reduced or prevented.

Reviewer 3 Report

Although I appreciated the Authors’ work, the presented review sounds like a hypothesis rather than a review supported by solid studies. It is well-known that EGCG has extremely poor bioavailability (decreases by extensive metabolism) and chemical instability (at 37°C, the stability of EGCG is 1.5 hours). Thus, much attention is paid to improving stability, release, and protection from extensive metabolic processes. Treatment of SARS-CoV-2 infections by tea consumption is not realistic. As it was nicely described by Henss et al. (work also cited in this review) “treatment of SARS-CoV-2 infections by oral tea consumption does not seem to be a realistic perspective. The consumption of two cups of green tea has been reported to result in a peak EGCG plasma level of <1 µM. However, as a small compound with broad antiviral activity, EGCG could potentially be used as a lead structure to develop highly effective antiviral drugs” (10.1099/jgv.0.001574). Therefore, the Authors’ statement that tea polyphenols regulate intestinal microecology to prevent and treat COVID-19 is greatly exaggerated.

Author Response

Responds to the reviewer's comments:

Dear Reviewers:

Thank you for your letter and for the reviewer's comments concerning our manuscript entitled “Tea polyphenols prevent and intervene in COVID-19 through intestinal microbiota” (ID: Foods-1560245). Those comments are very helpful for revising and improving our paper. We have studied comments carefully and have made a correction which we hope meet with approval. Revised portions are marked by “Track Changes”. The main corrections in the paper and the responses to the reviewer's comments are as follows:

Reviewer #3: Although I appreciated the Authors’ work, the presented review sounds like a hypothesis rather than a review supported by solid studies. It is well-known that EGCG has extremely poor bioavailability (decreases by extensive metabolism) and chemical instability (at 37°C, the stability of EGCG is 1.5 hours). Thus, much attention is paid to improving stability, release, and protection from extensive metabolic processes. Treatment of SARS-CoV-2 infections by tea consumption is not realistic. As it was nicely described by Henss et al. (work also cited in this review) “treatment of SARS-CoV-2 infections by oral tea consumption does not seem to be a realistic perspective. The consumption of two cups of green tea has been reported to result in a peak EGCG plasma level of <1 µM. However, as a small compound with broad antiviral activity, EGCG could potentially be used as a lead structure to develop highly effective antiviral drugs” (10.1099/jgv.0.001574). Therefore, the Authors’ statement that tea polyphenols regulate intestinal microecology to prevent and treat COVID-19 is greatly exaggerated.

Response: We appreciate your suggestions and recommendations. We have carefully referred to other relevant articles and made modifications in our paper. Our study is indeed a hypothesis, a new vision for a preventive approach to the future. We hope to provide some new perspectives for epidemic prevention through exploration in this direction.

Since the COVID-19 outbreak has been going on for two years, much of the research is focused on drug treatments and developing vaccines, which we have mentioned in this article. Our team is also actively exploring the impact of tea polyphenols on COVID-19. Of course, there is little literature to report that tea polyphenols can treat COVID-19, but there is much literature showing that tea polyphenols can treat inflammation, pneumonia or other viral diseases(We list them in the article). We provide some ideas for resistance to COVID-19 based on existing literature. 

At the same time, according to your suggestion, we replaced the words containing the treatment of COVID-19 in the article with prevention and improvement of the symptoms of COVID-19 to ensure the rigor of this article.

Round 2

Reviewer 1 Report

Authors fulfilled the asked questions.